# Adipose-Derived Exosomes as Possible Players in the Development of Insulin Resistance

**DOI:** 10.3390/ijms22147427

**Published:** 2021-07-11

**Authors:** Arkadiusz Żbikowski, Agnieszka Błachnio-Zabielska, Mauro Galli, Piotr Zabielski

**Affiliations:** 1Department of Medical Biology, Medical University of Bialystok, 15-089 Białystok, Poland; arkadiusz.zbikowski@umb.edu.pl (A.Ż.); mauro.galli@umb.edu.pl (M.G.); 2Department of Hygiene, Epidemiology and Metabolic Disorders, Medical University of Bialystok, 15-089 Białystok, Poland; agnieszka.blachnio@umb.edu.pl

**Keywords:** adipose tissue, metabolic disorders, insulin resistance, type 2 diabetes, adipokines, exosomes

## Abstract

Adipose tissue (AT) is an endocrine organ involved in the management of energy metabolism via secretion of adipokines, hormones, and recently described secretory microvesicles, i.e., exosomes. Exosomes are rich in possible biologically active factors such as proteins, lipids, and RNA. The secretory function of adipose tissue is affected by pathological processes. One of the most important of these is obesity, which triggers adipose tissue inflammation and adversely affects the release of beneficial adipokines. Both processes may lead to further AT dysfunction, contributing to changes in whole-body metabolism and, subsequently, to insulin resistance. According to recent data, changes within the production, release, and content of exosomes produced by AT may be essential to understand the role of adipose tissue in the development of metabolic disorders. In this review, we summarize actual knowledge about the possible role of AT-derived exosomes in the development of insulin resistance, highlighting methodological challenges and potential gains resulting from exosome studies.

## 1. Introduction

Adipose tissue (AT) is a multifunctional organ distributed across the human body in numerous locations. There are two main types of AT: brown adipose tissue (BAT) and white adipose tissue (WAT). Based on its localization, WAT can be divided into two major depots: subcutaneous adipose tissue (SAT) and visceral adipose tissue (VAT) [1]. Both tissue depots are endocrine organs, and their functional and different metabolic properties create a balance that contributes to the maintenance of energy homeostasis in the body [2]. This balance may be interrupted by obesity, a non-physiological WAT accumulation. The prevalence of obesity is continually on the rise in both developed and developing countries. The World Health Organization reported that, in 2016, more than 650 million adults were obese. Responsibility for this situation is primarily attributed to decreased physical activity and an increased intake of high-energy foods that are rich in saturated fats and sugars. Obesity may lead to numerous health comorbidities, such as cardiovascular disease, musculoskeletal disorders, cancer (including endometrial, prostate, breast, ovarian, liver, gallbladder, kidney, and colon), and, finally, insulin resistance (IR) and type 2 diabetes (T2D). Insulin resistance is a state in which insulin-responsive tissue (skeletal muscle, liver, adipose depots) fails to properly respond to physiological insulin levels, and it is strongly associated with obesity. Mechanisms linking IR with obesity are complex; however, low-grade chronic inflammation of adipose tissue and disturbances in the production and secretion of numerous adipokines are common denominators [3]. Endocrine functions of adipose tissue have been under continuous investigation since the 1980s, resulting in hundreds of potential bioactive secretory proteins and peptides [4]. Among these, leptin, adiponectin, resistin, and retinol-binding protein 4 (RBP4) have been studied extensively, which has led to the formulation of the definition of adipokines. Currently, adipokines are recognized as central regulators of insulin sensitivity, metabolism, and energy homeostasis. However, the full profile of adipose secretome—in the form of metabolites, proteins, and secretory vesicles—is still under continuous research [5,6].

A relatively new area of studies has since emerged, one that is dedicated to the exploration of WAT secretome, and more specifically to WAT-derived secretory extracellular vesicles (EVs). EVs are small membrane-derived vesicles secreted into extracellular space. Based on the differences in secretion mechanisms and the size of the particular EVs, secretory vesicles are divided into three main types: (1) apoptotic bodies, (2) microvesicles, and (3) exosomes. Apoptotic bodies, or blebs, with a diameter roughly above 1 µm, are released during the apoptosis of a cell [7]. Microvesicles are medium-sized structures (100 nm–1 µm) that originate from the plasma membrane through the process of blebbing. The microvesicle membrane contains surface antigens, receptors, and channels of the parent cell. The biogenesis and secretion of microvesicles is an ATP- and calcium-dependent process. Exosomes, in turn, display the smallest diameter (30–100 nm) and, contrary to the above EVs, they are released from the endosomal compartment of the cell in a process referred to as exocytosis. Briefly, cells generate exosomes through internal budding from the internal surface of intracellular lumen vesicles (ILVs) or endosomes. The fusion of exosome-containing vesicles with the plasma membrane results in exosome release [7]. Released exosomes travel with the blood to distant organs and interact with distant cells through a process of fusion, interacting via surface antigens and receptors or through internalization of the exosome by phagocytosis (Figure 1).

Exosomes are rich in proteins, glycoconjugates, lipids, DNA, and RNA molecules, all of which provide exosomes with a wide spectrum of potential effects on target cells [7]. Exosome–target cell interaction leads to the fusion of biological membranes and unloading of exosomal cargo, or to the triggering of intracellular signaling pathways via the activation of surface receptors. New data from proteome analysis of exosomes suggest that the composition of extracellular vesicles is richer than the classical WAT protein secretome, which offers a new perspective on the potential endocrine functions of WAT [8]. Conversely, circulating exosomes derived from various tissues may comprise a valuable source of information about the detrimental effects of obesity. Due to the central role of WAT in IR development, proteins, lipids, and RNA molecules from adipose tissue-derived exosomes may serve as potential biomarkers for IR and T2D. Moreover, RNAs present in adipose-derived exosomes may be directly involved in the regulation of the expression of target cell proteins, leading to the modification of signaling pathways and affecting the metabolism of distant tissues both positively and negatively. Due to many possibilities thatare intrinsic to the complex nature of exosomes, interest in this area of research is growing. However, research on tissue-derived exosomes provides numerous technicallimitations, which may lead to different observations and conclusions, despite the implementation of similar experimental models. Despite challenges, novel standardized approaches for exosomal isolation and identification allow for a better correlation between exosomal molecular composition and endocrine effects.

In this review, we aim to summarize the current findings regarding the exosomes produced by WAT by highlighting the possible functions of selected exosomal factors in the pathogenesis of insulin resistance and type 2 diabetes. Furthermore, the roles of the main cell populations of WAT in the secretory process will be pointed out, as will the possible use of exosomal content as a diagnostic and prognostic biomarker.

## 2. Current Understanding of WAT Cellular Composition and Secretory Function in Insulin Resistance

Continuous research has transformed our understanding of adipose tissue from an inert fat storage depot into a dynamic endocrine organ that modulates whole-body and individual tissue energy metabolism. Endocrine function of WAT depends on its localization, and it is affected by obesity and low-grade chronic inflammation connected with WAT hypertrophy and insulin resistance. The bulk of WAT resides in two major depots: subcutaneous adipose tissue (SAT) and visceral adipose tissue (VAT). SAT and VAT differ in size and metabolic activity regarding their respective adipocyte populations. SAT is the largest fat storage depot in the human body; therefore, it plays a crucial role in whole-body glucose and lipid management in response to insulin [9]. SAT adipocytes display a higher expression of proteins, typical for mature adipose cells, compared to VAT, which translates into a greater ability to accumulate intracellular TG inside a high number of larger lipid droplets. SAT is regarded as an expansion “buffer” that protects other tissues from adverse effects of ectopic lipid accumulation [10]. Smaller SAT cells are less active in the production of inflammatory factors such as interleukin-6, -8, and tumor necrosis factor (TNF) α [11], and are more responsive to insulin than larger VAT adipocytes [12]. Compared to subcutaneous fat depots, VAT is more sensitive to β-adrenergic lipolytic stimuli, and it displays greater responsiveness in FFA release under adrenergic stimulation [13]. Compared with SAT, VAT obesity is more likely to induce hepatic lipid accumulation and the development of hepatic IR [14] due to the direct portal vein connection of VAT depots to the liver [15]. Moreover, VAT adipocytes display a lower affinity to insulin at the level of the insulin receptor; therefore, they are less sensitive to insulin lipogenic stimuli than SAT adipocytes. The net result of above differences is that VAT displays a lower response to the lipogenic and anti-lipolytic effects of insulin and a higher sensitivity to β-adrenergic lipolytic stimulation [14,16]; thus, liver-related metabolic disorders are associated more with VAT enlargement than they are with SAT enlargement [17].

Adipocytes constitute the bulk of the cellular population of WAT. Non-adipocyte cells are commonly known as a WAT stromal vascular fraction (SVF) [18,19,20]. SVF comprises endothelial cells, preadipocytes, myeloid cells, fibroblasts, pericytes, macrophages, and adipose-derived stromal/stem cells (ADSCs). Both types of ADSCs, subcutaneous and visceral, have a mesenchymal origin and share similar properties, such as cell surface markers or cell viability. However, there is a difference regarding some key characteristics, such as motility, fatty acid metabolism, secretion, and inflammation [21]. ADSCs, together with adipocytes, are necessary in the process of adipogenesis, and therefore in the process of healthy lipid deposition [22]. Unfortunately, in high fat-diet-induced obesity and T2D, both the ADSCs population and anti-inflammatory action may be reduced. Moreover, ADSCs display an altered production and secretion of exosomes, which can lead to an increased probability of adipose tissue metabolic dysfunction [23,24,25]. Both the mature adipocytes and the various SVF cellular populations are a source of bioactive peptides and proteins, including cytokines, growth factors, chemokines, and hormones, which are described jointly as secretome. One of the first adipokines identified as a WAT secretome protein was leptin [26]. Leptin production occurs mostly in SAT and it shows a positive correlation with total body fat mass. Leptin exerts a positive effect on energy balance, mainly by acting on the brain, decreasing food intake, increasing energy expenditure, managing glucose and lipid metabolism, or altering neuroendocrine function. Obesity is often connected with leptin resistance [27], where elevated levels of leptin have a poor impact on metabolism [28,29]. Another important adipokine is adiponectin. Activation of adipoR1 and adipoR2 membrane receptors leads to an increase in FFA oxidation in liver and skeletal muscles [30]. Adiponectin has anti-inflammatory effects that contribute to a higher insulin response by adipose tissue. Obese individuals display lower adiponectin concentrations in the blood than their lean counterparts [31]. Adipose tissue also releases other metabolism-modulating proteins and peptides, such as resistin, RBP4, monocyte chemoattractant protein 1 (MCP-1), visfatin, chemerin, and vaspin. Newer candidates include WNT1-inducible-signaling pathway protein 2 (WISP2), or Metrnl [32,33]. All mentioned proteins act in the modulation of insulin sensitivity, and their levels are increased in cases of obesity and T2D [34]. Disturbance in the production of beneficial adipokines is associated with the induction of low-grade inflammation within WAT depots. This, in turn, correlates with the incidence of insulin resistance and T2D [35]. Adipose tissue macrophages release a set of pro-inflammatory proteins, such as interleukins (IL-1β and IL-6), TNFα, MCP-1, plasminogen-activated inhibitor (PAI-1), colony-stimulating factor (CSF), and inducible nitric oxide synthase (iNOS). The proinflammatory character of IL-1β, IL-6, and TNFα may lead to chronic inflammation and, as a consequence, to IR development in distant insulin-sensitive tissues [34]. The IL-1β can be regarded as a model cytokine that underlines the importance of macrophage-mediated inflammation in the induction of IR [36]. At the level of adipose tissue, obesity induces the release of IL-1β by the tissue macrophages, which, in turn, inhibits insulin signaling in adipocytes and stimulates the release of adipocyte pro-inflammatory adipokines [37,38]. The release of IL-1β by adipose tissue macrophages has significantly broader endocrine consequences, as adipose-derived IL-1β can directly induce IR in liver cells [39]. Moreover, insulin-resistant adipocytes mobilize TAG (triacylglycerol) stores, which, in turn, induce lipid-mediated insulin resistance in the liver in an IL-1β-dependent manner [40]. TNFα is another cytokine released by adipose tissue that is associated with the concept of tissue inflammation in cases of obesity [41]. TNFα takes action by two receptors, tumor necrosis factor receptor (TNFR) 1 and TNFR2, which further mediate JNK1 and IκB kinase/NFκB pathways. Both can directly inhibit insulin signaling by phosphorylating insulin receptor substrate 1 (IRS1), thus interrupting insulin action [42].

As adipokines and cytokines represent just a fragment of possible secretome sources of WAT, some studies have explored the association of adipose tissue secretome profiles with different metabolic conditions [43,44]. Apart from typical protein-based molecules and low-molecular weight metabolites, adipose tissue may modulate the metabolism of distant tissues through the production of extracellular vesicles (EVs). Compared to signaling factors or metabolites, EVs’ complex composition, which includes lipids, proteins, and nucleic acids, provides a number of opportunities to influence numerous processes, namely satiety, appetite, the body’s energy balance, glucose, and lipid metabolism.

## 3. The Role of Adipose Tissue-Derived Exosomes in Obesity and Insulin Resistance

WAT-derived exosomal cargo, which includes membrane-bound and soluble proteins, lipids, DNA and RNA sequences, small nuclear RNA (snRNA), transfer RNA (tRNA), and microRNA (miRNA), may induce both beneficial and negative effects on insulin sensitivity [45,46,47,48] (Table 1). Interestingly, the beneficial or negative exosome-mediated effects on the development of insulin resistance are strictly related to the type of exosome releasing cells. For instance, adipose tissue-derived exosomal miRNAs, miR-29a, miR-27a, miR-34, miR-155, and miR-223, affect metabolic processes through repression of PPARγ and IRS1, or M1 polarization of macrophages, resulting in insulin resistance of various tissues [46,49,50,51,52]. Contrary to the above effects, exosomes from adipose-derived mesenchymal stem cells alleviate the metabolic effects in a streptozocin-induced T1D model [53]. Due to the possible beneficial effects carried by certain exosome populations, the bioengineering and modification of exosomes and other EVs seem to be a promising approach for exosome-based target therapies. Unfortunately, processes underlying the incorporation into recipient cells of selective or unselective EVs remain elusive.

To fill gaps in the knowledge regarding the role of adipose tissue-derived exosomes in the development of obesity-induced insulin resistance, the identification of different WAT exosome sub-populations should be prioritized. Recently, several proteins, namely fatty acid-binding protein 4 (FABP4), adiponectin, and perilipin A, were identified and considered as markers for adipose-derived exosomes [54]. Proper identification of WAT-derived exosomes allows for a better description of their molecular cargo. Proteomic profiling of WAT secretome by Hartwig S. et al. revealed 817 “exoadipokines” in the form of proteins carried by exosomes. Among them, 67 proteins were not listed in the exosomal ExoCarta exoadipokines database (http://www.exocarta.org/, accessed 30 May 2018), meaning they constituted novel members among WAT-derived exosomes. Among them, the authors linked SERPINF1, COL5A3, COL1A1, C3, RBP4, GPD1, and TUBA1C to endocrine system disorders, metabolic diseases, and type 2 diabetes [8].

Adipose-derived exosomes are also crucial for the physiological function of WAT stores. Dai M. et al. showed that adipose-derived exosomes may promote adipogenesis via ADSCs stimulation in both in vivo and in vitro models, suggesting a key role of exosomes in adipose tissue regeneration [55]. The process of healthy adipogenesis is associated with the expansion of WAT; thus, proper vascularization for new cells is needed. ADSCs-derived exosomes promote vascularization by the secretion of various miRNAs such as miRNA-21, which improves angiogenesis by increasing the expression of hypoxia-inducible factor 1 (HIF-1α), Akt, ERK, and SDF-1 [56] in endothelial cells in an in vitro model. Moreover, in the presence of vascular endothelial growth factor C (VEGF-C), ADSCs produce exosomes that are rich in miRNA-132, which leads to intensified lympho-angiogenesis by regulating TGF-β/Smad signaling in lymphatic endothelial cells in in vitro models [57].

Apart from the adipogenesis process, healthy WAT plays a role in immunoregulation within WAT tissue. The population of adipose tissue macrophages (ATMs) is a crucial regulator of local and systemic inflammatory responses. Depending on the polarization of macrophages, the inflammation process may be strengthened or weakened; therefore, two main subtypes of macrophages—M1 and M2—were identified. M1 macrophages are responsible for inflammatory signaling, while the M2 subtype displays anti-inflammatory properties that participate in the attenuation of the inflammatory response. M2 macrophages release anti-inflammatory cytokines such as IL-10 [58]. Various mechanisms are involved in the control of ATM polarization, and one of them is identified as an exosome-dependent mechanism. WAT cells, such as mature adipocytes or ADSCs, can reprogram ATMs from the pro-inflammatory M1 profile to the anti-inflammatory M2 profile, or vice versa. Adipocyte-derived exosomes rich in miRNA-34a promote the M1 profile [50], while ADSCs-derived exosomes rich in STAT3 protein promote the M2 profile [25] (Figure 2). Interestingly, M2 macrophages, in turn, produce exosomes enriched with miRNA-690; these improve insulin sensitivity in high-fat diet mouse models [45] (Figure 2). Moreover, miRNA exosomal cargo modulates inflammatory processes within adipose tissue. A study by Flaherty et al. suggests that exosomal lipid content and composition may also have an impact on inflammatory regulation within WAT [59].

The exosomal route of intercellular communication has also been identified between different adipose stores. A study by Zhang et al. showed that WAT-derived exosomes rich in miRNA-210/92a enhance FGFR-1 expression in BAT [61]. On the other hand, BAT is also an active player in organ cross-talking through exosome production. Mouse models fed a high-fat diet, and which therefore suffered from obesity and insulin resistance, showed that BAT-derived exosomes isolated from healthy animals may improve metabolic function in their obese counterparts. BAT-derived exosomes decreased the weight of obese animals, lowered blood glucose levels, and improved glucose tolerance. Moreover, lipid accumulation in WAT and the liver decreased after BAT-derived exosome application; however, the molecular mechanisms underlying these processes are still elusive [62]. One mechanism may be associated with miR-132-3p and its hepatic target, Srebf1. MicroRNA miR-132-3p, released in BAT-derived exosomes, leads to suppression of Srebf1, which is involved in lipogenesis in the liver [63]. Similar findings were reported by Gao et al. in relation to WAT-derived vesicles, where exosomes from lean mice attenuated appetite and reduced body weight in obese mice, and vice versa [64]. The appetite-reducing effect was attributed to the direct impact of exosomes on the hypothalamus via the mTOR signaling pathway. This finding suggests that exosomes may be central to investigating new potential anti-obesity strategies that involve both adipose tissue and the hypothalamus.

Due to strong connections with the induction of obesity, WAT-derived exosomes are also a prime suspect in the induction of insulin resistance and the promotion of type 2 diabetes. Moreover, exosomes could be a promising source of novel diagnostic biomarkers in T2D [54]. It has been observed that overall adipose exosome secretion is increased in IR [65]. Adipocytes collected and cultured from rats and mice demonstrate increased exosome secretion upon incubation with fatty acids, induction of oxidative stress, or stimulation with antidiabetic drug glimepiride [66]. Adipose tissue exosomes activate macrophages in a TLR-4 dependent manner and with RBP4 involvement. They lead to an increased migration of macrophages, the production of pro-inflammatory cytokines, such as IL-6 or TNFα, and are consequential for IR development [67]. It was found that miRNA-223, as well as miRNA-34a, may be involved in the modulation of macrophages activity [50,51]. miRNA-34 changes the polarization of ATMs into pro-inflammatory M1 phenotypes by targeting transcription factor KLF4, and its effect increases mutually by degree of obesity [27]. Furthermore, after migration, ATMs produce exosomes with miRNA that may enhance insulin signaling and lead to IR; for instance, miRNA-155 can influence adipocyte differentiation by repressing PPARγ [47]. Moreover, as mentioned before, the miRNA-155 found in exosomes secreted by adipocytes evokes an additional proinflammatory effect on macrophages by activating STAT1 and repressing STAT6 expression [52]. Moreover, obese adipose tissue macrophages secrete exosomes rich in miRNA-29, leading to IR via PPARδ targeting [46]. All of the above findings suggest that exosomal signaling is involved in the induction of insulin resistance via the promotion of inflammation.

Beyond the induction of inflammation, exosomes can directly modulate the insulin sensitivity of liver and skeletal muscles through miRNA or protein-dependent mechanisms. A study by Li et al. showed that elevated circulating levels of exosome-bound miRNA-222 observed in HFD-induced obese mice originating from gonadal WAT, and are responsible for the induction of skeletal muscle and liver IR via the repression of IRS-1and phosphor-AKT levels [48]. Another mechanism may also be associated with the decrease in exosomal miRNA-141-3 levels. Obese adipose tissue-derived exosomes transfer far less miRNA-141-3 into hepatocytes, which leads to inhibition of hepatic glucose uptake and IR [60]. Exosomes may also play a crucial role in the pathogenesis of cardiovascular disorders, which are common comorbidities of obesity. For instance, an increased level of lncRNA-NBR2 in exosomes may lead to heart muscle collagen deposition and decreased cardiac function. Moreover, the obese pregnant mice model suggests that placenta inflammation and fetal cardiac dysfunction may be associated with decreased miRNA-19b in VAT-derived exosomes [68]. Animal models also suggest that perivascular adipose tissue-derived exosomes promote the migration of vascular smooth muscle cells (VSMC) and lead to diabetes-induced vascular dysfunction [69].

The correlation of WAT-specific exosomes with an induction in insulin resistance in distant tissues was also observed in the case of protein composition. In the bloodstream, exosomes reach the liver and muscle tissue, triggering glucose intolerance and IR [47]. For instance, a recent study shows that high-fat diet mice produce more CD-36 fatty acids that transport protein-containing exosomes in WAT which, after internalization by hepatocytes, lead to hepatic lipid accumulation and inflammation [70]. The above data show obese adipose tissue-derived exosomes in a bad light; however, WAT also contains an ADSCs population with anti-inflammatory properties. In recent years, these cells have been of specific interest regarding the positive effects of ADSCs-derived exosome administration on metabolic disorders. It has been shown that, in the presence of pro-inflammatory cytokines, ADSCs secrete exosomes with immunosuppressive properties [71]. ADSCs-derived exosomes contain immunomodulatory factors, such as TGFβ, IL-6, IL-8 IL-10, CCL2, or hepatocyte growth factor (HGF) [72]. Moreover, it has been shown that ADSCs-derived exosomes possess an active STAT3 protein, which may improve insulin sensitivity and glucose tolerance. Additionally, STAT3 may promote a polarization switch of macrophages from an M1 type to an anti-inflammatory M2 type, subsequently decreasing tissue inflammation [25]. ADSCs-derived exosomes can also affect lymphocyte populations; for instance, exosomal miRNA-10a promotes differentiation of Th17 and Treg from naïve CD4+ T cells, which is crucial in the regulation of immune response in tumors and inflammatory disease [73].

## 4. Exosomes as Potential Biomarkers of Insulin Resistance and Type 2 Diabetes

The changes in the content and composition of exosomes may also serve as both diagnostic and prognostic markers of metabolic disturbances. Exosomes, due to their endosomal origin, are rich in biomolecules that reflect the condition of cells. Moreover, easily accessible body fluids such as urine, blood, or saliva contain exosomes produced by various cell types, including WAT, creating a considerable opportunity for the discovery of novel and easily accessible biomarkers [7,74]. Regarding glucose-related metabolic disturbances, WAT-derived exosomes contain numerous proteins involved in insulin signaling, such as phospho-AKT, phospho-p70S6K, phospho-S6RP, phospho-insulin receptor, phosphor-IGF1R, leptin receptor, and FGF21. In T2D, the levels of phospho-insulin receptor and leptin receptor are decreased in exosomes [65]. Proteomic analysis of adipose-derived exosomes, collected from obese rats, showed higher levels of caveolin 1, LPL, and aquaporin 7 in comparison to control animals [75]. In contrast, levels of mitochondrial enzyme AK2, catalase, and carboxylesterase were lower than those in the control group [75]. Additionally, adipocyte-derived exosomes are characterized by a decreased ratio of adiponectin to total protein content [76]. Furthermore, obesity modifies the miRNA pattern in exosomes released by adipocytes. Recent works show that levels of miRNA-130b and miRNA-27a increased in exosomes. miRNA-130b is positively correlated with the degree of obesity, and it may play important role in adipose tissue-muscle crosstalking by TGFβ signaling, whereas miRNA-27a may induce IR in skeleton muscle via PPARγ repression [49,77]. Obesity and insulin resistance significantly affect WAT-derived exosomes, which could be utilized as a diagnostic and/or prognostic marker of metabolic disturbances. Recent studies indicate that exosomes produced by adipocytes from insulin-resistant subjects contain proteins that are strongly associated with insulin resistance and T2D, such as calreticulin, S100A6, mimecan, PARK7/DJ1, PPIB, and tenascin [78]. One of the more promising biomarkers of WAT-induced insulin resistance could be perilipin A. In the model of murine obesity, perilipin A was described as a biomarker of AT-derived exosomes that increase in number during the progression of obesity. Both perilipin A and exosome levels correlated positively with insulin resistance [79]. Another potential exosomal biomarker present in the course of T2D is cystatin C, which positively correlates with metabolic complications of obesity such as myocardial infarction, vascular disease mortality, subsequent vascular events, and metabolic syndrome [80,81]. Similarly, exosomal CD 14 protein content is positively correlated with cardiovascular events and negatively correlated with dyslipidemia and decreased risk of T2D development [80,81]. It seems that both markers, cystatin C and CD14, may be useful tools in the assessment of comorbidity in T2D patients.

Exosomal cargo in the form of miRNA or proteins possesses all the necessary features to be a high-quality biomarker of IR/T2D and attendant comorbidities. Analysis of blood exosomes is a non-invasive way to observe treatment effects and disease progression. Unluckily, recent knowledge about exosomal content is still too fragmentary to allow the implementation of exosomal proteins in diagnosis. Moreover, new biomarkers need strong proof of sensitivity and specificity, and the results of an exosomal examination should be reproducible. Unfortunately, there is no gold standard for exosome collection, purification, and analysis, leading to significant variability.

## 5. Methodological Challenges in the Study of Adipose-Derived Exosomes

WAT-derived exosomes represent a promising field of research; however, its development is currently constrained by methodological limitations. Although numerous approaches for exosomal isolation are available, there is no golden standard in this first step of analysis. The most common method is ultracentrifugation [82]; however, this technique has several disadvantages, such as possible non-exosomal vesicular contaminations, low reproducibility, low yield, potential damage to exosomes, and the overall low-throughput of centrifugation-based isolation techniques [83]. A comparative study of the isolation methods shows significant variability in exosomal miRNA content [84,85]. Moreover, proteomic analysis also shows significant variability in exosomal preparations. It seems that the biggest challenge in the proteomic characterization of exosomal composition, at least in the case of proteomic-focused studies, is contamination with typical high-abundance plasma proteins. For instance, albumin, immunoglobulins, and metalloproteinases are common co-isolated plasma proteins [86] that may “hide” low-abundance exosome proteins during the mass-spectrometry analysis of protein composition [87].

Additionally, exosome-related studies employing an in vitro cellular approach requires careful execution, as stability in culture conditions is important to yield high-quality exosomes. For example, too low/high glucose levels and antibiotics lead to changes in cell exosome secretion and content [88,89]. Reports show that bacteria such as mycoplasma may interfere with the results of exosome examination by production of their own extracellular vesicles [90,91]. Moreover, exosome content may change with the passage number and the age of cells used for exosome production [92,93]. Furthermore, too high a seeding density may decrease the number of exosomes per cell and increase the overall non-exosomal protein contamination of the medium [94]. Cell culture media is also a potential interfering factor in exosome studies. One of the most crucial media components, fetal bovine serum (FBS), is a direct source of exosomes that are isolated during serum production. It was reported that FBS-derived exosomes may be co-isolated from medium, which leads to problems with proper cell-derived exosome analysis. Unfortunately, there is no standardized protocol for FBS-derived exosome depletion. One approach is to culture cells in FBS-free media, thus solving this complication. However, on the other hand, this approach condemns cells to growth factor starvation, which may also have an impact on cell exosome production and content [95]. Another proposition regarding the depletion of FBS-derived exosomes is the use of filtration or ultracentrifugation, as well as the use of commercial exosome-depleted FBS. However, these processes may also remove important growth factors from FBS [96].

Further purification of isolated exosomes can improve reproducibility by reducing non-exosomal cellular debris and other contaminants, such as unbound proteins. Size-exclusion chromatography (SEC) [94] and asymmetric-flow field-flow fractionation [97] are promising methods that yield subpopulations of exosomes with different biophysical and molecular properties. Further characterization of subpopulations can be realized by the morphology-based identification of exosomes (transmission electron microscopy, scanning electron microscopy, cryo-electron microscopy, or atomic force microscopy), size-based identification (nanoparticle tracking analysis and dynamic light scattering), or surface marker detection (Western blotting, trypsin digestion, flow cytometry, mass spectrometry, and ELISA analysis) [98]. Since all exosomes originate from endosomes, characteristic antigens should be confirmed on their surface. Tetraspannins (CD9, CD82, CD81, and CD63) or proteins involved in MVB formation (TSG101, Alix) are suitable for the confirmation of the exosomal origin of vesicles [99].

Improvement in exosome isolation and analysis is an essential step that will lead to a better understanding of the exosomal mechanism of cellular crosstalk and the creation of new exosome-based medical applications.

## 6. Summary

Our current understanding of adipose tissue physiology and its involvement in metabolic disturbances has changed dramatically in recent times. Currently, adipose tissue secretome studies are the focus of numerous research groups. Within this field, descriptions of exosome secretion and signaling seem to be central for a better understanding of adipose tissue functionality and its impact on the metabolism of individual insulin-sensitive tissues and the body as a whole. Exosomes may be a proverbial treasure trove containing knowledge about new mechanisms of crosstalk within the human body, as well as a great source of potential disease biomarkers and new therapeutical strategies for the control of obesity, insulin resistance, and T2D.

## 7. Conclusions

Adipose tissue-derived exosomes are an emerging player in the field of obesity-induced insulin resistance and type 2 diabetes.WAT-derived exosomes impact the metabolism of distant tissue through a multitude of bioactive molecules, including proteins, lipids and RNAs.Exosomes from healthy subjects can improve insulin sensitivity in their obese counterparts. Application of specific exosomes may act as a promising treatment for IR and T2D.Reproducible isolation and identification of particular exosomes from various biological matrices is challenging. Emerging methodologies significantly improve exosomal preparations, leading the way to future clinical applications.

## Figures and Tables

**Figure 1 ijms-22-07427-f001:**
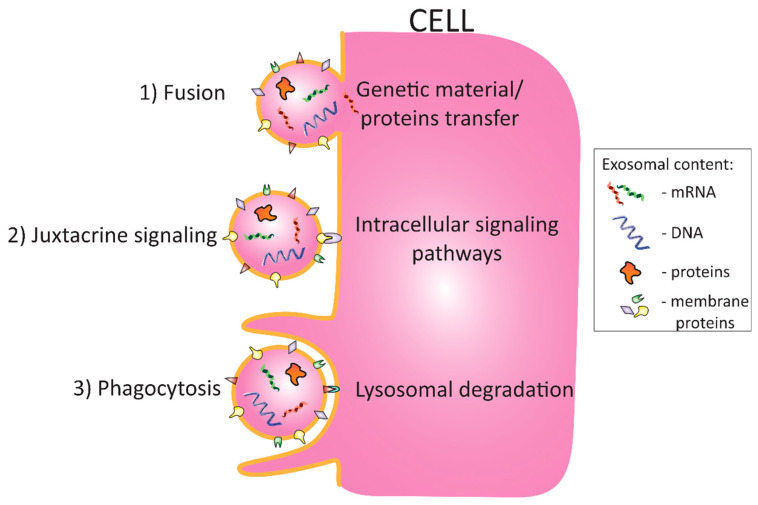
Exosome-target cell interaction: (1) membrane fusion allows for the transfer of both the internal cargo molecules and membrane proteins by merging with the plasma membrane of a target cell; (2) junxtacrine signaling activates intracellular signaling pathways through ligation of target cell membrane receptors with exosome membrane ligands; (3) phagocytosis allows for the internalization of the whole exosome by target cell. It allows for the lysosomal degradation of exosomes. The biological effects of exosomes’ action depend on a type of interaction with the target cell [7].

**Figure 2 ijms-22-07427-f002:**
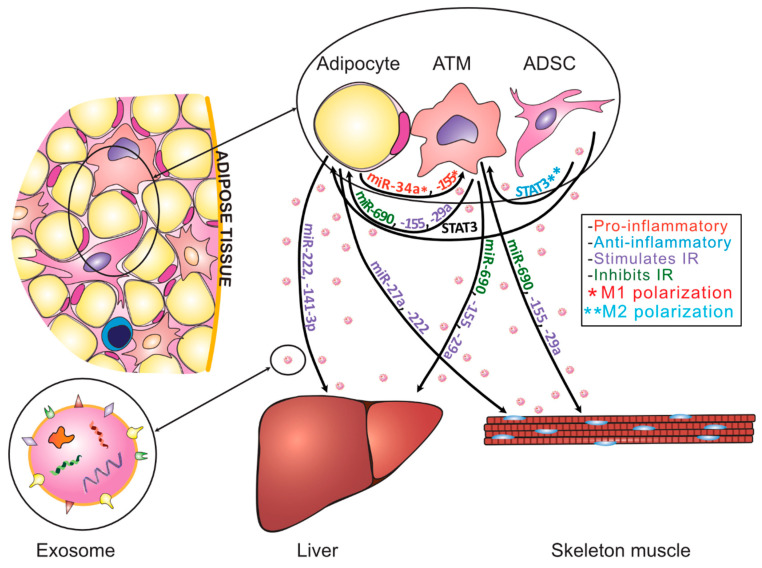
Graphical summary of adipocytes, ATMs, and ADSCs-derived exosome composition and their targets. The color of the factor indicates its overall effect. Red: miR-34a [50] and miR-155 [52] are produced by obese adipocytes and lead to M1 polarization of macrophages and tissue inflammation. Blue: STAT3 is produced by ADSCs and inhibits tissue inflammation by induction of M2 polarization of macrophages [25]. Purple: miR-27a [49], -29a [46], -155 [47], and -222 [48] are released by obese adipocytes and M1 ATMs, and induce IR in insulin sensitive tissues. Downregulation of exosomal miR-141-3p release by obese adipocytes induces insulin resistance in liver [60]. They are responsible for the induction of IR in WAT, liver, and skeletal muscle by decreasing the glucose uptake or impairing tissue-specific insulin signaling. Green: miR-690 is produced by M2 macrophages and increases insulin-stimulated glucose uptake and enhances insulin-mediated suppression of glucose output; increases in insulin-stimulated AKT phosphorylation [45]. Black colored STAT3 stimulates transformation of WAT into beige fat tissue, improving its metabolic function [25]. IR—insulin resistance; *M1 polarization of macrophages; **M2 polarization of macrophages.

**Table 1 ijms-22-07427-t001:** Crosstalk between adipose-derived exosomes and target cells. The table describes exosome experimental models, emphasizing possible factors and their biological effects on the target tissue.

Model	Parent Cell/Tissue	ExosomalFactor	Target Cell/Tissue	Outcome	Reference
In vitro	Lean human WAT derived ADSCs	miRNA-132	Human dermal lymphatic endothelial cells	Increase in lymphoangiogensisby regulating TGF-β/Smad signaling	Wang, X. et al. [57]
In vivo	Mice WAT (mainly adipocytes of obese epididymal WAT)	Overexpressed miR-34a	Macrophages	Inhibition of M2 profile polarization and tissue inflammation	Pan, Y. et al. [50]
In vitro/in vivo	Mice epididymal fat ADSCs	STAT3	Macrophages/WAT	Inhibition of macrophages inflammatory response by M2 polarization (in vitro)/ beiging of WAT (in vivo)	Zhao, H. et al. [25]
In vitro	ADSCs isolated from inguinal fat pad of 3-week-old Lewis male rats	Overexpressed miR-21	HUVEC cells	Promotion of vascularization	An, Y. et al. [56]
In vitro	M2 macrophages	miR-690	3T3-L1 adipocytes and L6 myocytes; HFD mice hepatocytes	Increase in insulin-stimulated glucose uptake in both 3T3 adipocytes and L6 myocytes; enhanced insulin-mediated suppression of glucagon-stimulated glucose output in isolated primary hepatocytes; increase in insulin-stimulated AKT phosphorylation in all cell types	Ying, W. et al. [45]
In vitro	SVF of obese VAT	miR-223	Primary mice macrophages	Regulation of both pro- and anti-inflammatory pathways, including the E3 ubiquitin ligase FBXW7 (hypothesis)	Deiuliis, J. A. et al. [51]
In vitro/ in vivo	ATMs isolated from obese mice VAT	Overexpressed miR-155	3T3-L1 adipocytes and L6 muscle cells and primary hepatocytes; adipocytes and hepatocytes in vivo	Reduction in insulin-stimulated glucose uptake in 3T3-L1 adipocytes and L6 muscle cells; decrease in expression of the miR-155 target gene PPARγ and decrease in expression of GLUT4; impair the suppressive effect of insulin on glucose production in hepatocytes	Ying, W. et al. [47]
In vitro	Obese mice adipocytes	miR-155	Macrophages	Induction of M1 polarization	Zhang, Y. et al. [52]
In vitro/ in vivo	Obese mice ATMs of VAT	miR-29a	3T3-L1 adipocytes, L6 myocytes, and primary hepatocytes	Inhibition of glucose uptake in adipocytes and myocytes, and promotion of glucose output in hepatocytes; the IR signaling is abrogated by miR-29a	Liu, T. et al. [46]
In vitro	Obese mice adipocytes	miR-27a	C2C12 skeletal muscle cells	Impairment of insulin signaling via repression of PPARγ	Yu, Y. et al. [49]
In vivo	Obese patients’ gonadal WAT	miR-222	Liver, skeletal muscle	Impairment of insulin sensitivity by repressing IRS1	Li, D. et al. [48]
In vitro	Obese mice WAT	Downregulated miR-141-3p	AML12 hepatocytes	Inhibitory effects on insulin sensitivity and glucose uptake of AML12	Dang, S. et al. [60]

## Data Availability

Not applicable

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
