# Peer review of "Adipose-Derived Exosomes as Possible Players in the Development of Insulin Resistance"

_ijms, 2021, doi:10.3390/ijms22147427_

Round 1
Reviewer 1 Report
In this manuscript, Zbikowski et al first reviewed the heterogeneity and multiple functions of white adipose tissue, discussed the functions of adipose tissue-derived exosomes, and then overviewed the challenges for exosome isolation. The topic of the manuscript is interesting. However, the issues listed below have damped the overall enthusiasm.
- Although the title indicates the focus of this review article would be adipose-derived exosomes, the authors used the majority of the manuscript to discuss the heterogeneity and different functions of white adipose tissue, which have been extensively reviewed elsewhere and diminish the novelty of this article.
- The contents are poorly organized. For example, the authors started to discuss three types of EVs in Line 238-247. All of a sudden, they began to discuss the uptake of exosomes in the recipient cells in lines 248-249. In lines 273-282, they jumped back to discuss different types of EVs again. For the section “4. Exosomes”, a lot of information seems randomly put together. It is very difficult to understand the logic and structure of this section.
- Figure 1 seems not very relevant to the focus of this review article. The schematic in Figure 2 is difficult to understand and misses one important function of exosomes --- promote the macrophage polarization. Table 2 is not cited anywhere in the manuscript and seems randomly included in the paper.
- After reading all the problems raised in “Section 5. Methodological challenges”, it is difficult to trust the results summarized in section 4. What kind of results can we trust and what kind of results shall we take cautions? Could the authors comment on this?
Minor points:
- Language editing and proofreading are highly recommended. Just a few examples below: Line 38 “thought”, not sure what it means in the sentence; Line 73 “RNA particles”; Line 163 “ADSCs fraction”; Line 454 “bacterias”; Line 466 “by themself”
- Please give a full name when an abbreviation is introduced for the first time.
- In section 4, please double-check which terms (exosomes or endosomes) should be used.
- There is only one table. Not sure why it is listed as “Table 2”
Author Response
Reviewer #1
In this manuscript, Zbikowski et al first reviewed the heterogeneity and multiple functions of white adipose tissue, discussed the functions of adipose tissue-derived exosomes, and then overviewed the challenges for exosome isolation. The topic of the manuscript is interesting. However, the issues listed below have damped the overall enthusiasm.
Reply: We are grateful for the thorough review of our manuscript and the time spent by the Reviewer to improve our work. We extensively re-written large parts of the manuscript, shortened or removed unnecessary information and introduced new paragraphs. We hope that the changes introduced in the manuscript will earn the Reviewer recommendation for the publication of our manuscript. We would be grateful for any other comments by the Reviewer regarding the further improvement of the paper. As the changes introduced in our manuscript were numerous, their indication in text would make the reading uncomfortable. We supply 2 versions of the manuscript. Revised, and PDF version with the changes highlighted via track changes feature.
1. Although the title indicates the focus of this review article would be adipose-derived exosomes, the authors used the majority of the manuscript to discuss the heterogeneity and different functions of white adipose tissue, which have been extensively reviewed elsewhere and diminish the novelty of this article.
Reply: According to the Reviewer suggestions we extensively re-written the manuscript to put more emphasis on the role of WAT exosomes in the induction of metabolic disturbances connected with obesity. Former paragraph 2 (“Understanding WAT tissue heterogeneity”) and paragraph 3 (“WAT as a multifunctional secretory organ”), were compacted and shortened. We retained the basic information on the heterogeneity and secretory activity of adipose tissue in the combined short paragraph 2 entitled “Current understanding of WAT cellular composition and secretory function in insulin resistance”. We opted not to remove this basic information altogether, as we reference various types of adipose tissue cells and secretory adipokines numerous times later in the manuscript. This short paragraph will also give some perspective on the current state of adipose tissue secretome and how it can be compared with exosome-related secretome which is extensively described later in the manuscript. Moreover, we extended the data on the possible use of WAT-derived exosomes as diagnostic biomarkers of insulin resistance and type 2 diabetes. This subject is described in the separate paragraph 4, entitled “Exosomes as potential biomarkers in insulin resistance and type 2 diabetes”.
2. The contents are poorly organized. For example, the authors started to discuss three types of EVs in Line 238-247. All of a sudden, they began to discuss the uptake of exosomes in the recipient cells in lines 248-249. In lines 273-282, they jumped back to discuss different types of EVs again. For the section “4. Exosomes”, a lot of information seems randomly put together. It is very difficult to understand the logic and structure of this section.
Reply: We are grateful for pointing us toward those insistencies within our manuscript. We thoroughly re-organized section 4 for greater clarity (now section 3 entitled “The role of adipose tissue-derived exosomes in obesity and insulin resistance”. Some of the information regarding the types of the secretory vesicles was moved to Introduction. We hope that the changes will significantly improve the overall clarity of the manuscript.
3. Figure 1 seems not very relevant to the focus of this review article. The schematic in Figure 2 is difficult to understand and misses one important function of exosomes --- promote the macrophage polarization. Table 2 is not cited anywhere in the manuscript and seems randomly included in the paper.
Reply: Figure 1 was composed in the hope to introduce the reader to the topic of secretory vesicles and their interaction with target cells. We agree that this basic information should not be presented later in the manuscript. We opted to move the Figure 1 and its description to the matching part of the Introduction section. Regarding the Figure 2, we added information about the polarization of the macrophages as requested by the Reviewer. We hope that the extensive description of the Figure 2 contents in the figure caption will help the reader to navigate the information presented in the figure.
4. After reading all the problems raised in “Section 5. Methodological challenges”, it is difficult to trust the results summarized in section 4. What kind of results can we trust and what kind of results shall we take cautions? Could the authors comment on this?
Reply: The extended section 5 now includes additional discussion on the possible improvements of exosomal isolation and characterization. Referenced works by various research teams employ methodologies which significantly reduce uncertainties surrounding the quality of exosomal preparations and prove their exosomal origin.
Minor points:
1. Language editing and proofreading are highly recommended. Just a few examples below: Line 38 “thought”, not sure what it means in the sentence; Line 73 “RNA particles”; Line 163 “ADSCs fraction”; Line 454 “bacterias”; Line 466 “by themself”
Reply: We extensively edited and proof-red the whole manuscript. We would be grateful for the comment if the language needs to be corrected further.
2. Please give a full name when an abbreviation is introduced for the first time.
Reply: Corrected.
3. In section 4, please double-check which terms (exosomes or endosomes) should be used
Reply: Checked.
4. There is only one table. Not sure why it is listed as “Table 2”
Reply: Thank you for pointing towards this mistake. Reference to Table 2 was removed.
Reviewer 2 Report
This is excellent ...well written, excellent presentation and very good for teaching...
Only suggestion would be for you to have 4 most important bullet points for the fast reader...
Author Response
Reviewer #2
This is excellent ...well written, excellent presentation and very good for teaching...
Only suggestion would be for you to have 4 most important bullet points for the fast reader...
Reply: We would like to thank the reviewer for the valuable comments. As suggested by the Reviewer we introduced 4 bullet points in the revised Summary section. We hope that the changes introduced in the manuscript will earn the Reviewer recommendation for the publication of our manuscript.
Reviewer 3 Report
This review article deals with an important and novel aspect of adipose tissue physiology, the role of exosomes in normal functions and in obesity and insulin resistance. The authors discuss in detail these topics and the material will be useful to the readers of the IJMS. However the manuscript could be easier to read if the authors present the main topic of the review, the functions and defects of the exosomes at the initial section of the manuscript. As it is, the manuscript has a lengthy presentation of well known aspects of adipose tissue biology and pathology, leaving the main topic, the exosomes, for the final sections.
In addition the manuscript will benefit with a comprehensive revision of the English language use and sintaxi.
Abstract: Adipose tissue (AT) is an endocrine organ involved in the management of whole-body 9 and individual organ energy metabolism via secretion of adipokines, hormones and secretory micro 10 vesicles. The secretory process occurs through direct release of the bioactive molecules into the 11 bloodstream and through the secretion of small membrane-derived vesicles called exosomes, which 12 are rich in biologically active factors such as proteins, lipids or RNA. AT function may be affected 13 by pathological processes which occur within adipose depots. One of the most important is obesity, 14 which triggers adipose tissue inflammation and adversely affects release of beneficial adipokines. 15 Both processes may lead to further AT dysfunction, contributing to the changes in the whole-body 16 metabolism and subsequently to insulin resistance. According to recent data, changes within pro- 17 duction, release, and content of exosomes produced by AT may be essential to understand the role 18 of tissue in the development of metabolic disorders. In this review, we summarize actual knowledge 19about the possible role of AT-derived exosomes in insulin resistance development, highlighting 20 methodological challenges and potential gains resulting from exosomes studies. 21
Citation: Arkadiusz Żbikowski;
Blachnio-Zabielska, A.; Galli, M.;
Author Response
Reviewer #3
This review article deals with an important and novel aspect of adipose tissue physiology, the role of exosomes in normal functions and in obesity and insulin resistance. The authors discuss in detail these topics and the material will be useful to the readers of the IJMS. However the manuscript could be easier to read if the authors present the main topic of the review, the functions and defects of the exosomes at the initial section of the manuscript. As it is, the manuscript has a lengthy presentation of well known aspects of adipose tissue biology and pathology, leaving the main topic, the exosomes, for the final sections.
Reply: We are grateful for the effort put by the Reviewer into the improvement of our manuscript. According to the Reviewer suggestions, we re-organized the introduction section and significantly shortened the lengthy presentation of adipose tissue biology and pathology. We retained some basic information from the paragraph 2 and 3 in a new paragraph 2 entitled “Current understanding of WAT cellular composition and secretory function in insulin resistance”, as we reference various types of adipose tissue cells and secretory adipokines numerous times later in the manuscript. Moreover, we extended the data on the possible use of WAT-derived exosomes as diagnostic biomarkers of insulin resistance and type 2 diabetes. This subject is described in the separate new paragraph 4, entitled “Exosomes as potential biomarkers in insulin resistance and type 2 diabetes”. We hope that the changes introduced in the manuscript will earn the Reviewer recommendation for the publication of our manuscript. We would be grateful for any other comments by the Reviewer regarding the further improvement of the paper.
In addition the manuscript will benefit with a comprehensive revision of the English language use and sintaxi.
Reply: We extensively edited and proof-red the whole manuscript. We would be grateful for the comment if the language needs to be corrected further. The changes introduced in our manuscript were numerous, and their in-text indication would make the reading uncomfortable. We supply 2 versions of the manuscript. Revised, and PDF version with the changes highlighted via track changes feature.
Round 2
Reviewer 1 Report
I appreciate all the efforts and time the authors invested to massively revise the manuscript. I like the revised layout much better. Now it is really a review with a focus on exosomes.
Some minor points:
Figure 1 figure legend: typo - please change endosomes to exosomes
Line 465: authors seemed to forget one of the most important cytokines from WAT --- IL-1b
Line 1221, 1314: another typo - please change microsomes to exosomes
I still spot quite a lot of grammar errors. The authors need to find a good language editor to go over the manuscript one more time. Below are just a few examples:
L608: macrophages subtypes
L609: M2 are possess
L1189: BAT exosomes derived
L1365: in obese
L1372: IR adipocytes
L1565: RNA should be RNAs
.....
Author Response
Comments and Suggestions for Authors
I appreciate all the efforts and time the authors invested to massively revise the manuscript. I like the revised layout much better. Now it is really a review with a focus on exosomes.
Response: We are grateful for the additional comments by the Reviewer and for the in-depth evaluation of our work. We introduced all the additional changes in the manuscript suggested by the Reviewer. Thank you for pointing us to several grammar/language mistakes which were overlooked. Moreover, to definitively solve the issue of poor language, we approached professional manuscript proof-reading service to help us with the language correction. The “track changes” corrected version is submitted as PDF supplement file alongside the revised manuscript. We hope that the final version of our work will gain the Reviewer recommendation.
Sincerely,
Piotr Zabielski, PhD
Some minor points:
Reply: The manuscript was corrected in all the suggested places. Thank you. Moreover, we submitted the manuscript for the external proof-reading service to improve the overall language of our work. The “track-changes” version of the manuscript in PFD format is submitted as supplement file for the review purpose.
Figure 1 figure legend: typo - please change endosomes to exosomes - Corrected
Line 465: authors seemed to forget one of the most important cytokines from WAT --- IL-1b
Reply: The information about this important cytokine was indeed missing. The paragraph “Current understanding of WAT cellular composition and secretory function in insulin resistance” now includes additional information (L164-L174), which underlines the importance of this cytokine in the macrophage-mediated paracrine and endocrine signaling connected with insulin resistance.
Line 1221, 1314: another typo - please change microsomes to exosomes - Corrected
I still spot quite a lot of grammar errors. The authors need to find a good language editor to go over the manuscript one more time. Below are just a few examples:
L608: macrophages subtypes - Corrected
L609: M2 are possess - Corrected
L1189: BAT exosomes derived - Corrected
L1365: in obese - Corrected
L1372: IR adipocytes - Corrected
L1565: RNA should be RNAs - Corrected
Reviewer 3 Report
The authors have introduced the changes suggested and the MS has been much improved
Author Response
Thank you for the evaluation of our work. To further improve its quality, we approached professional manuscript proof-reading service (external) to help us with the language correction. The “track changes” corrected version is submitted as PDF supplement file alongside the final version of the manuscript the manuscript.
Sincerely,
Piotr Zabielski, PhD